# Real-World Data on Severe Cutaneous Adverse Reactions to Drugs

**DOI:** 10.3390/ph19010021

**Published:** 2025-12-22

**Authors:** Sergey Zyryanov, Elizaveta Terehina, Olga Butranova, Irina Asetskaya, Vitaly Polivanov, Alexander Yudin

**Affiliations:** 1Department of General and Clinical Pharmacology, Peoples’ Friendship University of Russia Named After Patrice Lumumba (RUDN University), 6 Miklukho-Maklaya St., 117198 Moscow, Russia; zyryanov-sk@rudn.ru (S.Z.); butranova-oi@rudn.ru (O.B.); asetskaya-il@rudn.ru (I.A.); 2Pharmacovigilance Center, Information and Methodological Center for Expert Evaluation, Record and Analysis of Circulation of Medical Products Under the Federal Service for Surveillance in Healthcare, 4-1 Slavyanskaya Square, 109074 Moscow, Russia; pvit74@gmail.com; 3Moscow City Health Department, City Clinical Hospital No. 24, State Budgetary Institution of Healthcare of the City of Moscow, Pistzovaya Srt. 10, 127015 Moscow, Russia; youdine@gmail.com; 4Department of Immunology, Russian National Research Medical University Named After N.I. Pirogov, St. Ostrovityanova, 1, 117997 Moscow, Russia

**Keywords:** severe cutaneous adverse reactions (SCARs), pharmacovigilance, drug reaction with eosinophilia and systemic symptoms (DRESS), Stevens–Johnson syndrome (SJS), toxic epidermal necrolysis (TEN), linagliptin, piperacillin, antibacterial agents

## Abstract

**Background/Objectives**: Cutaneous adverse drug reactions (CADRs) represent the most common manifestations of drug-induced allergy, with most unfavorable clinical outcomes seen in severe cutaneous adverse reactions (SCARs). To manage SCARs immediate cessation of the offending drug is needed; therefore, it is crucial to identify the list of medications associated with SCARs in real-world clinical practice. The objective of this study was to evaluate the structure of drugs associated with SCARs and to analyze drug-induced SCAR signals by calculating the reporting odds ratio (ROR) and proportional reporting ratio (PRR) based on spontaneous reports extracted from the Russian national pharmacovigilance database. **Methods**: A retrospective, descriptive pharmacoepidemiological analysis of spontaneous reports (SRs) registered in the pharmacovigilance database from 1 April 2019 to 31 March 2025. **Results**: A total of 7011 SRs with SCARs were finally revealed, with 907 identified drug triggers. The most frequently reported were antibacterial drugs for systemic use (22.8%), antineoplastic agents (17.8%), and antiepileptics (6.0%). The top five drugs involved in SCARs were dupilumab (2.14%, *n* = 244), piperacillin and beta-lactamase inhibitor (2.0%, *n* = 227), pembrolizumab (1.98%, *n* = 225), levofloxacin (1.95%, *n* = 222), and linagliptin (1.93%, *n* = 220). The strongest signals were detected for linagliptin (PRR = 15.37, 95% CI: 13.54–17.44; ROR = 17.24, 95% CI: 14.95–19.88), followed by clindamycin (PRR = 12.44, 95% CI: 10.89–14.21; ROR = 13.62, 95% CI: 11.77–15.77) and by piperacillin and beta-lactamase inhibitor (PRR = 10.02, 95% CI: 8.86–11.43; ROR = 10.81, 95% CI: 9.42–12.40). **Conclusions**: Pharmacovigilance databases facilitate the identification of diverse phenotypes of SCARs and the list of culprit drugs. The accumulated data serve as a valuable tool to enhance clinical practice outcomes and strengthen overall healthcare monitoring.

## 1. Introduction

Drug safety remains a pressing issue worldwide, with unpredictable reactions manifested as drug allergy among the most prevalent. Skin is a typical target organ involved in various drug-induced allergic reactions (Gell and Coombs hypersensitivity types I–IV). Most studies reported the incidence of CADRs in hospitalized patients to fall within the range of 1% to 5% [1,2]. A higher incidence was observed in critically ill patients, reaching 11.6% [3].

The majority of CADRs are usually mild and may resolve rapidly without the need for additional interventions; however, in certain instances, SCARs can occur, necessitating hospitalization and often posing a potentially life-threatening risk to patients [4]. The global incidence of SCARs ranges from 0.4 to 1.2 cases per million individuals annually, with a mortality rate between 14% and 70% [5]. Most common SCARs include drug reaction with eosinophilia and systemic symptoms (DRESS), Stevens–Johnson syndrome (SJS), toxic epidermal necrolysis (TEN), and acute generalized exanthematous pustulosis (AGEP). TEN is characterized by the highest lethality rate, followed by SJS and AGEP [6,7]. A nationwide, register-based matched cohort study performed in Denmark revealed that patients with TEN have a high rate of both short-term and long-term mortality. An average of 19.3% of patients die within 30 days and up to 28% within 90 days. Comparing median survival for patients with and without TEN, authors reported the values of 4.77 years (95% credit interval, CI: 2.02–9.46) and 25.96 years (95% CI: 25.19–27.20), respectively. It is of particular interest that even two years after diagnosis, the mortality risk in TEN patients compared to controls increased by 230% (HR 3.30; 95% CI: 2.34–4.66) [8].

It has been established that virtually any drug, regardless of dosage form and route of administration, can cause SCAR development. The spectrum of drugs involved in SCARs is broad; published data indicate antimicrobials and antiepileptics as the top culprit classes [9,10,11], though many other drugs have been shown to cause various types of SCARs, such as antifungals [12], anti-gout agents [13], monoclonal antibodies [14], antineoplastics [9,15], and vaccines [16].

Genetic predisposition constitutes a primary risk factor for the development of SCARs. There is strong evidence of the involvement of several genes, mainly human leukocyte antigens (HLA), in altered immune signaling, resulting in SCARs [17,18]. Other genetic markers of SCARs may include some genotypes of drug metabolism enzymes, drug transporters, and T-cell receptors [19]. Most studies demonstrated a higher overall incidence of SCARs in females, especially in case of SJS and TEN [20]. A meta-analysis conducted by Fathima, S. et al. (2024) identified male gender and HIV infection as primary risk factors for SJS/TEN; however, the analysis ultimately included only five studies, and the findings were deemed to be of low quality [21].

Diagnostics of SCARs should be based on medial history, anamnesis, clinical manifestations, and time of their development. The time interval for SJS and TEN is 4–28 days after the start of exposure, for AGEP 1–12 days, for DRESS 2–8 weeks, and for maculopapular drug exanthem 4–14 days [22]. Accurate identification of the causative agent and subsequent discontinuation of its administration represent the most critical steps in the management of SCARs. This process poses a considerable challenge for clinicians, as most patients are simultaneously subjected to multiple pharmacological interventions. To effectively address this clinical challenge, it is recommended to rely on lists of already-identified SCAR triggers, particularly those with a strong drug–adverse drug reaction (ADR) causal relationship. This information comes from both classical clinical trials and studies conducted within pharmacovigilance systems.

At present, diverse methodologies are employed to monitor drug safety within contemporary pharmacovigilance practices. Among these, the analysis of national and supranational databases of SRs of ADRs has demonstrated notable efficacy. Key advantages of this approach include the capacity to detect rare and severe ADRs, and identify pharmacological triggers of certain ADRs during the post-marketing phase. Additionally, it facilitates the assessment of causality between ADRs and drugs and provides comprehensive data on drug utilization across all patient populations in real-world clinical settings over extended durations, with dynamic updates captured in real time [23,24].

The aim of our study was to assess the structure of drugs involved in SCARs and analyze signals of drug-induced SCARs, calculating the ROR and PRR using SRs extracted from the Russian national pharmacovigilance database.

## 2. Results

### 2.1. General Characteristics of SMQ SCAR

The total number of SRs with preferred terms (PTs) related to the standardized Medical Dictionary for Drug Regulatory Activities (MedDRA) query (SMQ) “SCAR” was 12,475 (6-year study period: from 1 April 2019 to 31 March 2025). After excluding duplicates and invalid messages, 7011 SRs were included in the study. Table 1 contains data on the structure of PTs within SMQ SCAR and demographic characteristics of patients derived from SRs. The mean age of patients according to the data derived from SRs with SMQ SCARs was 55.7 ± 22.4 years (min = 6 days, max = 99 years), 50.9% (*n* = 3570) were males, 44.5% (*n* = 3120) females, and in 4.6% (*n* = 321) the gender was unknown.

The most prevalent PT revealed in our study was DRESS (*n*= 1272), followed by toxic skin eruption, TSE (*n* = 1134), TEN (*n* = 803), erythema multiforme, EM (*n* = 709), and SJS (*n*= 697). In further analysis, we focused on the top 10 PTs.

A predominance of male gender was detected in most of PTs, but there were some exceptions where females were leaders, such as for dermatitis bullous (DB; males—45.7%, females—51.9%), dermatitis exfoliative (DE; males—35.7%, females—55.8%), and dermatitis exfoliative generalized (DEG; males—41.7%, females—52.1%).

The lowest mean age of patients was revealed in SRs with target skin lesion (TSL) (43.2 ± 25.6 years), while in SRs with oculomucocutaneous syndrome the highest mean value was reported (71.3 ± 13.2 years). The age distribution for the total SCAR sample and for top 10 PTs is presented in Table 2. According to SR data, SCARs developed mainly in young and middle-aged persons (18–59 years), the age group of 60–74 years ranked second (25.9%), and the age group of 75–89 years ranked third (16.6%) in the overall structure. It is worth noting that 53 cases (0.8%) of SCARs developed in neonates.

Analysis of annual dynamics of SCAR reporting was carried out for the period of 2020–2024; since in the year 2019 information was available only for 9 months (*n* = 101) and in 2025 for 3 months (*n* = 549). A stable increase in the number of SRs for SMQ SCAR was revealed over the observation period, with nearly a five-fold increase from 2020 to 2024 (Figure 1A). Considering the dynamics of reporting among the top 10 PTs, the same trends were observed for all except skin necrosis (61 SRs in 2023 versus 52 in 2024; Figure 1B). Detailed data on the total number of SRs for each PT are presented in Appendix A.

Estimating the country of SR origin revealed a total of 96 countries. The top five countries according to SCAR reporting included France (22.1%, *n* = 1560), Russia (14.1%, *n* = 985), Japan (8.6%, *n* = 603), USA (6.2%, *n* = 428), and China (6.1%, *n* = 417). Data on the distribution of SRs by country of origin are given in Table 3.

Analysis of the reporters indicated that the majority of SRs with SCARs (85.2%, *n* = 5974) were submitted by employees of pharmaceutical companies, while a significantly smaller proportion originated from healthcare professionals (10.0%, *n* = 701). Regional pharmacovigilance centers, authorized regulatory bodies, and personnel from World Health Organization (WHO) Centers for International Drug Monitoring were also involved in reporting, but with a drastically smaller contribution. Detailed information on reporter categories is provided in Table 4.

The outcomes identified in SRs demonstrated a predominance of generally favorable patterns. In the total sample of SRs with SCARs (*n* = 7011), 35% of SRs resulted in recovery without sequelae (*n* = 2457) and 26.8% in improvement in condition (*n* = 1880). Death was reported in 5.8% (*n* = 407) of the overall sample. Among the top 10 PTs, recovery without sequelae and improvement in condition were the most frequently reported outcomes. The highest proportion of lethal outcomes was revealed for TEN (28%), SJS (11.9%), and DRESS (4.2%). The outcome distribution for the total sample of SRs with SMQ SCAR and for the top 10 PTs is presented in Figure 2. Detailed data for all PTs detected are given in Appendix A.

Application of seriousness criteria led to the classification of 99.3% (*n* = 6964) of SRs as serious. The distribution of SRs by these criteria is presented in Table 5. The most frequently revealed seriousness criterion was “important medical events” (*n* = 3110, 44.4%), followed by “hospitalization or its prolongation” (*n* = 2761, 39.6%) and death (*n* = 407, 5.8%).

### 2.2. Suspected Drugs

One SR included ≥1 suspected drug, and a total of 11,701 drugs were identified in 7011 SRs with SCARs. After excluding drugs not classified by the Anatomical Therapeutic Chemical (ATC) classification, 11,376 drugs were detected. The total list of drugs for each PT is presented in Appendix A. To generate a list of possible drug triggers of SCARs, each drug was counted once; the final list included 907 suspected drugs (identified by their unique international non-proprietary name (INN)).

We revealed 14 ATC first-level groups involved in SMQ SCAR (Figure 3). The highest frequencies were detected for the following groups: J—antiinfectives for systemic use (*n* = 3442; 30.3%), L—antineoplastic and immunomodulating agents (*n* = 2677; 23.5%), and N—drug for treatment of the nervous system (*n* = 1509; 13.3%). Appendix A demonstrates full data on the ATC first-level groups revealed in SCARs.

The ATC groups (first level) revealed for the top 10 PTs are demonstrated in Figure 4, and detailed data regarding all PTs are given in Appendix A. Antiinfectives for systemic use (group J) ranked first in the structure of suspected drugs in DRESS (42.8%), TSE (38.1%), TEN (27.5%), and AGEP (37.4%). Antineoplastic and immunomodulating agents (group L) were the leaders in EM (35.8%), SJS (34.4%), DEG (31.4%), cutaneous vasculitis (CV; 34.1%), and skin necrosis (SN; 47.7%). ATC group A (alimentary tract and metabolism) was the most prevalent in SRs with DB (29.3%).

The next step was to classify 907 identified drug triggers into ATC second-level groups to identify the most frequently reported pharmacological groups involved in SCARs. Figure 5 illustrates five main pharmacological classes revealed for SMQ SCAR and for the top 10 PTs. In the total sample of SRs with SCARs, the following ATC second-level groups were the most frequently reported: J01—antibacterial drugs for systemic use (22.8%), L01—antineoplastic agents (17.8%), N03—antiepileptics (6.0%), B01—antithrombotic agents (5.3%), and L04—immunosuppressants (4.5%). Analyzing the leading ATC second-level groups among the top 10 PTs, we revealed the L01 group to head at least half of the cases (TEN, EM, SJS, CV, SN). The J01 group was the leader in DRESS, TSE, and AGEP, while the A10 group ranked first in DB and D11 in DEG (Figure 5).

The subsequent phase of the analysis involved identifying the top five drugs exhibiting the highest reporting frequencies at the SMQ and the PT levels. Among the entire SCAR group, the following drugs were leaders: dupilumab (2.14%, *n* = 244), piperacillin and beta-lactamase inhibitor (2.0%, *n* = 227), pembrolizumab (1.98%, *n* = 225), levofloxacin (1.95%, *n* = 222), and linagliptin (1.93%, *n* = 220). Data on the top five drugs for the top 10 PTs are presented in Figure 6. Information on the top 10 drugs for the top 10 PTs is presented in Appendix A.

### 2.3. Disproportionality Analysis

The next phase of our study was aimed at safety signal detection. A disproportionality analysis was conducted to identify drug-adverse event (AE) pairs exhibiting statistically elevated reporting frequencies for SCARs in association with the specified drug. The analysis employed signal detection methodologies commonly utilized in pharmacovigilance, including the computation of the ROR and PRR values, as detailed in the Section 4. PRR and ROR values, along with their corresponding 95% confidence intervals (CIs), were calculated for each of the top 10 drugs with the highest reporting proportions at the SMQ SCAR level.

Disproportionality analysis performed with PRR calculation revealed notable safety signals for SCARs across several agents. Linagliptin demonstrated the strongest association (PRR = 15.37, 95% CI: 13.54–17.44), followed by clindamycin (PRR = 12.44, 95% CI: 10.89–14.21) and the combination of piperacillin with beta-lactamase inhibitor (PRR = 10.07, 95% CI: 8.86–11.43). Detailed data regarding the results of the PRR analysis performed for the top 10 drugs with the highest reporting proportions at the SMQ SCAR level are presented in Table 6 and Appendix A.

ROR analysis for the same top 10 drugs revealed considerable variation in point estimates. Linagliptin exhibited the highest ROR (17.24, 95% CI: 14.95–19.88), followed by clindamycin (13.62, 95% CI: 11.77–15.77) and piperacillin and beta-lactamase inhibitor (10.81, 95% CI: 9.42–12.40). Detailed results of the ROR analysis for the top 10 drugs with the highest reporting frequencies at the SMQ SCAR level are given in Table 7.

## 3. Discussion

We analyzed 7011 SRs with SCARs submitted to the national Pharmacovigilance database over a 6-year period (February 2019–March 2024). These SRs originated from 96 countries, with 57.1% of cases occurring in France, Russia, Japan, USA, and China. The proportion of SRs marked as serious was 99.3%, with 407 cases (5.8%) resulting in death. There were 11,376 culprit drugs revealed in 7011 SRs. Our analysis allowed us to identify 907 INNs suspected in SCAR development. Th main SCAR phenotypes in our study were DRESS (*n* = 1272), TEN (*n*= 803), SJS (*n* = 697), and AGEP (*n* = 486).

In our study of SRs with SCARs, a negligible predominance of males was detected (50.9%), with the highest proportion of males in epidermal necrosis, TSL, and AGEP (65.3%, 61.5%, and 59.3%, respectively). Female gender was more frequently detected in DE and DEG (55.8% and 52.1%). Published data present conflicting evidence regarding the impact of patient gender on the development of SCARs. Some works consider female gender a risk factor of SCARs [3,25,26]. Keller, S.F. et al. (2018) stated that female gender is associated with a 2.5-fold increase in allopurinol-induced SCARs compared with male gender [25]. This observation is of particular interest, as allopurinol is used primarily in men, who are more likely to suffer from hyperuricemia and gout. Women have been consistently observed to utilize healthcare services more frequently and, as a result, receive a higher number of prescriptions. This increased exposure may theoretically contribute to a marginally elevated overall risk of experiencing ADRs, including SCARs.

Our study revealed the following age distribution of SRs with SCARs: 7.4% were from pediatric population, 34.0% from young and middle-aged adults, and 44.3% from persons > 60 years. Age is not considered to be a significant direct risk factor, but individuals belonging to older age groups tend to be more susceptible to developing SCARs, as they are exposed to many medications due to aging and multimorbidity [27,28]. It is known that drug allergy is less common in children, but current data challenge the general idea that the incidence of SCARs in children is lower than in adults [29,30]. Although SCARs can occur in children of any age, even in neonates, a systematic review by Afiouni, R. et al. (2021) indicated a mean age of onset of pediatric SCARs to be in the limits of between 8 and 10 years, particularly for DRESS [31]. Another review underscored the significance of DRESS among pediatric SCARs, with particular emphasis on the association between antibiotic-induced DRESS and the subsequent onset of autoimmune diseases [32].

We found that the most frequently reported ATC first-level groups involved in SCAR development were J—antiinfectives for systemic use (30.3%), L—antineoplastic and immunomodulating agents (23.5%), and N—nervous system (13.3%). Our findings exhibited comparable patterns of suspected medications to those reported by Li, D. et al., who conducted an analysis of 77,789 SRs with SCARs over a span of 17 years (1 January 2004 to 31 December 2021) within the FAERS database and revealed antibacterials (20.6%), antiepileptics (16.7%), and antineoplastics (11.3%) as being among the leading culprits in pharmacological classes [9]. A Korean study of individual case safety reports (*n* = 755) indicated antiepileptics (19.5%) and antibiotics for systemic use (12.7%) as the classes most involved in SCAR development [33]. Another study from South Korea, utilizing a nationwide multicenter registry spanning from 2010 to 2015 across 34 hospitals, identified a significant role of antibacterial agents in the development of SCARs, together with allopurinol and carbamazepine [34]. Antiepileptics are known to be common causes of SCARs like SJS and TEN [35]. The most involved antiepileptics demonstrated in published data are lamotrigine, valproic acid, carbamazepine, levetiracetam, and phenytoin [36,37,38]. Carbamazepine ranked seventh among the top 10 agents involved in SCARs in our study (*n* = 183, 1.61%), with significant signal detection (PRR = 8.64, 95% CI: 7.50–9.96; ROR = 9.17, 95% CI: 7.88–10.68), which is in line with a huge volume of published data describing the role of antiepileptics in SCARs.

Antineoplastic agents employed in cancer chemotherapy are well recognized for inducing a variety of toxic effects, and contemporary research increasingly substantiates their contributory role in SCAR development [9,15,39]. The antineoplastics identified as culprit drugs in our study were mainly monoclonal antibodies, and consideration of their role is provided in subsequent sections of the discussion.

Estimating the total structure of drugs involved in SCARs, we revealed that half of the top 10 drugs (Appendix A) were antibacterials and 3 drugs were monoclonal antibodies. The top five medications involved in SCARs were dupilumab (2.14%, *n* = 244), piperacillin and beta-lactamase inhibitor (2.0%, *n* = 227), pembrolizumab (1.98%, *n* = 225), levofloxacin (1.95%, *n* = 222), and linagliptin (1.93%, *n* = 220).

A retrospective study (a 10-year period was analyzed) of antibacterials involved in SCARs in China [10] as well as a Taiwanese study [40] indicated cephalosporines as the main cause of CADRs (*n* = 20, 31.7% and *n* = 17, 23.0%, respectively), followed by penicillins (*n* = 16, 25.4% and *n* = 20, 30.0%) and quinolones (*n* = 12, 19.0% and *n* = 5, 6.8%). Russian data revealed third-generation cephalosporins as leaders among antibacterials involved in SJS and TEN (32%) [11]. It is important to highlight that beta-lactams are consistently identified as primary agents implicated in SCARs, as demonstrated by multiple pharmacovigilance studies [11,33,34,41,42]. Our analysis demonstrated that piperacillin combined with a beta-lactamase inhibitor (*n* = 227, 2.0%) ranked second among the drugs implicated in SCARs. Zhang, H. et al. (2024) ranked piperacillin third among the drugs involved in SCARs [43]. Genetic predisposition was demonstrated for some beta-lactams, including piperacillin [44] and amoxicillin. For the last one, a drug association with *HLA-B*15:01* [odds ratio (OR) = 22.9, 95% CI: 1.68–1275.67; *p* = 7.34 × 10^−3^] was demonstrated [45]. A study of genetic associations between beta-lactams and SCARs in native Thai subjects revealed significant risks for carriers of alleles *HLA-A*01:01*, *HLA-B*50:01*, *HLA-C*06:02*, *HLA-DRB1*15:01*, *HLA-DQA1*03:01*, and *HLA-DQB1*03:02* [46].

Piperacillin and amoxicillin were detected among the top 10 drugs involved in SCARS in our study, with formulations of both agents comprising combinations with beta-lactamase inhibitors. Rivolta, F. et al. assumed that beta-lactamase inhibitors such as clavulanic acid may significantly contribute to the development of CADRs, with patients exhibiting exclusively mucocutaneous reactions demonstrating greater sensitization to clavulanic acid compared to the β-lactam antibiotic itself [47].

Levofloxacin ranked fourth in our study (*n* = 222, 1.95%). A characteristic feature of SCARs induced by fluoroquinolones is the high mortality rate compared to most other groups of drugs [34]. Levofloxacin was shown to be involved in CADRs like fixed drug eruptions [48]. In the study of SJS and TEN triggers in Russia, levofloxacin accounted for up to 4% in the structure of antibacterials involved in the named SCARs [11].

Clindamycin is not listed among top SCAR triggers according to published data; however, there are some cases of clindamycin-induced CADRs described by authors from different countries [49,50,51,52,53]. Our results revealed clindamycin to be among the top 10 drugs involved in SCARs (*n* = 203, 1.78%), and disproportionality analysis revealed a strong signal detection (PRR = 12.44, 95% CI: 10.89–14.21; ROR = 13.62, 95% CI: 11.77–15.77).

An important issue highlighted in our study was the significant contribution of monoclonal antibodies (MAbs) to the development of SCARs. Pembrolizumab, nivolumab, and dupilumab were revealed to be among the top 10 drugs involved in SCARs. Pembrolizumab ranked third in the overall list of culprit drugs (1.98%, *n* = 225), and it was the leader in SRs with SJS and TEN. Nivolumab ranked ninth in the total structure of culprit drugs revealed in SRs with SCARs (1.41%, *n* = 160). Pembrolizumab and nivolumab are immune checkpoint inhibitors used in cancer treatment. To date, sufficient data have been accumulated indicating an association between MAbs and the development of SCARs, especially SJS and TEN. Zhu, J. et al., analyzing the FAERS database from 2004 to 2020, revealed 253 cases of SJS and 184 cases of TEN associated with immune checkpoint inhibitors (SJS/TEN: ROR = 2.88, 95% CI: 2.61–3.17). Pembrolizumab was found to be a culprit drug in 183 cases of SJS/TEN (SJS/TEN: ROR = 4.93; SJS: ROR = 4.36; TEN: ROR = 5.49) and nivolumab in 175 cases of SJS/TEN (SJS/TEN: ROR = 2.43; SJS: ROR = 2.29; TEN: ROR = 2.21) [54]. Dupilumab ranked first in the total structure of drugs involved in SCARs in our study (*n* = 244, 2.14%); the greatest contribution was made by DEG (here dupilumab also headed the drug list, with 19.77%, *n* = 85) and by EM (5.81% of all drugs involved in EM, *n* = 58). Dupilumab is a MAb that selectively binds to the alpha subunit of interleukin-4 receptor (IL-4Rα) and blocks the effects of IL-4 and IL-13. Its indications include a broad spectrum of allergic diseases, such as asthma, eosinophilic esophagitis, chronic spontaneous urticaria, etc. Among novel spheres of dupilumab use, it is important to note the treatment of immune-mediated CADRs induced by immune checkpoint inhibitors [55]. Dupilumab is known to produce various hypersensitivity reactions (e.g., anaphylaxis, AGEP, EM, serum sickness or serum sickness-like reactions, angioedema, generalized urticaria, rash, erythema nodosum) [56], though the most common ADRs seen in real-world practice include nasopharyngitis, conjunctivitis, upper respiratory tract infections, injection site reactions, alopecia, vitiligo, and psoriasis [57].

Disproportionality analysis performed in our study indicated linagliptin as the drug with the most significant safety signal for SCARs (PRR = 15.37, 95% CI: 13.54–17.44; ROR = 17.24, 95% CI: 14.95–19.88). Linagliptin ranked fifth in the total list of drugs involved in SCARS based on our results, and it was the first on the list of drugs involved in DB (*n* = 206, 23.09%). Huang, J. et al. (2020) [58] revealed disproportionality signals regarding SCARs for dipeptidyl-peptidase-4 (DPP-4) inhibitors based on FAERS database analysis (ROR = 6.12, 95% CI 4.69–8.00). The authors found ROR values indicating significant signal disproportionality for linagliptin (5.51, 95% CI 2.72–11.19), saxagliptin (4.84, 2.14–10.91), sitagliptin (5.34, 4.01–7.13), and vildagliptin (6.30, 2.33–17.01) [58]. Another study utilizing data from VigiBase^®^ indicated a significant bullous pemphigoid signal for the group of DPP-4 inhibitors (ROR = 179.48, 95% CI: 166.41–193.58). The strongest signal was detected for teneligliptin (ROR = 975.04, 95% CI: 801.70–1185.87) [59]. ROR calculation for bullous pemphigoid performed within the FAERS database indicated the group of DPP-4 inhibitors to be the group with the highest association with mentioned SCARs: Comparison to all other drugs in the FAERS resulted in a ROR value of 109.79 (95% CI: 101.61–118.62) and comparison to other diabetes medications resulted in a ROR value of 74.46 (95% CI: 60.58–91.52) [60]. Published data point out high rates of DPP-4 inhibitor involvement in CADR development, including various hypersensitivity reactions [61,62,63]. The mechanisms of SCAR induction by DPP-4 inhibitors are still unrevealed, though excessive apoptosis of keratinocytes may be considered, which is supported by the data obtained by Duraisamy, P. et al. (2022), who found necrotic keratinocytes in 44% of histological samples of patients with CADRs induced by DPP-4 inhibitors [64].

Finally, we need to make an emphasis on the limitations of our study that may impact the interpretation and reliability of our findings. Despite the method of SRs representing the cornerstone of the postmarketing surveillance of drug safety, the nature of SRs may lead to underreporting and selective reporting bias, limiting representativeness and comprehensive safety profiling. The most salient limitation lies in the fact that the cumulative number of SRs does not correspond to the actual consumption rates of a given drug in the population, thereby precluding the estimation of true ADR incidence. It should be added that the retrospective character of our analysis, combined with the potential for incomplete data indicated in SRs, limits the ability to accurately predict the risk factors associated with the development of SCARs. Another important concern is related to disproportionality analysis (ROR and PRR calculation). Disproportionality analysis is aimed at the identification of patterns of disproportionate reporting of specific AEs in patients exposed to a particular drug, relative to overall reporting rates within the pharmacovigilance database. However, these statistical disproportions do not establish direct causal relationships. The observed strength of a signal may arise from alternative explanations summarized in the work by Fusaroli, M. et al. (2025) [65]: reverse causality (the drug was used to treat the event after diagnosis), protopathic bias (the drug was used to address early symptoms of an undiagnosed event), indication bias (underlying condition prompting drug use increases susceptibility to the event), notoriety bias (heightened regulatory or media attention enhances reporting), selective reporting practices, and variability among reporters.

## 4. Materials and Methods

The process of AE collection in the Russian Federation is governed by the central pharmacovigilance authority, the Federal Service for Surveillance in Healthcare (Roszdravnadzor), through the Automatized Information System (AIS) “pharmacovigilance” database, whose functioning is organized in compliance with the ICH E2B (R3) standard [66] and in integration with WHO and Naranjo algorithms. Identification of ADRs within the AIS is carried out utilizing MedDRA version 28.0 [67]. Reports submission to the AIS is carried out by healthcare professionals, pharmaceutical industry personnel, patients, and their representatives. The AIS database was used as a data source in our study. The most widely used pharmacovigilance databases are the World Health Organization’s (WHO’s) VigiBase^®^, the European Medicines Agency’s EudraVigilance, and the FDA’s FAERS databases. However, any national pharmacovigilance database may significantly enhance our understanding of real-world drug safety, giving valuable insights into ethnicity-specific ADR profiles. Thus, national pharmacovigilance databases constitute a valuable tool of postmarketing drug safety, providing a comprehensive overview of ADR characteristics. In this study, our research team presents an extensive characterization of SRs obtained from the Roszdravnadzor AIS database, facilitating an in-depth analysis of SCARs.

The design of our study is a retrospective, descriptive pharmacoepidemiological analysis of SRs registered in the AIS database from 1 April 2019 to 31 March 2025 (a total study period of 6 years). Causality assessment of ADRs was conducted using the Naranjo algorithm, with only those classified as “certain,” “probable,” or “possible” included in the final dataset. The Naranjo algorithm is a validated and structured method used for over 40 years in routine pharmacovigilance to assess the causal relationship between a suspected medicinal product and ADR. Currently, the Naranjo Algorithm is considered the “gold standard” for the initial assessment of ADR cases in clinical trials and is one of the most well-known and frequently used algorithms worldwide, allowing for the comparison of data from different authors and studies. Its advantages include standardization (common criteria, reproducibility), objectivity (minimizes subjectivity and variability in the assessment of the ADR–drug causal relationship), and simplicity of use. Limitations include low sensitivity in complex and non-standard cases. In such instances, the issue of a causal relationship was reviewed collegially by clinicians and expert staff of the pharmacovigilance center (at least 3 different specialists). Inclusion of ADRs marked with a “certain,” “probable,” or “possible” causal relationship in the final safety analysis is a standard of competent pharmacovigilance. It acknowledges the probabilistic and population-based nature of postmarketing surveillance. While individual SRs marked with a “possible” causal relationship are weak evidence (since they could be explained by disease or other drugs), in aggregate they form the essential epidemiological substrate for early signal detection, safeguarding public health by ensuring the safety surveillance system is tuned for sensitivity.

Drug identification in the SRs identified in our study was carried out using the ATC classification. For inclusion, SRs originating from Russia or other countries had to contain PTs coded as SCARs. Within an SMQ, PTs can be categorized as either narrow or broad. Narrow-scope PTs denote terms that are more specifically indicative of the condition or focal area of interest, whereas broad-scope PTs may ultimately exhibit limited or negligible relevance for inclusion in the analysis [68]. There are 21 “narrow” PTs within SMQ SCAR in MedDRA version 28.0 (Table 8).

If one of the “narrow” PTs listed in Table 8 was found in an SR in the “adverse reaction” field, we included that SR in the study. Duplicates and invalid SRs were excluded. The validation procedure was carried out in accordance with paragraph 407 of the “Rules of Good Pharmacovigilance Practice of the EAEU” [69], according to which SRs must contain four essential elements: an identifiable reporter, an identifiable patient, at least one suspected medicinal product, and at least one suspected ADR. SRs that lacked one or more of these elements were considered invalid. Completeness of the information pertaining to the suspected drug, the ADR, and patient and reporter data were estimated by each author. The seriousness of the ADR was assessed based on data presented in paragraph 2 of the EAEU guidelines [69]. A flowchart of our study is given in Figure 7.

Suspected medications were initially identified by their INNs and subsequently classified according to the ATC first- and second-level groups. Demographic data were extracted from SRs, and all data processing was carried out using Microsoft Excel 2019. A descriptive statistical approach was applied; qualitative variables were summarized by their absolute and relative frequencies (*n*, %), while quantitative variables were described using the mean and standard deviation. Statistical significance testing was not performed, as the analysis of SRs is characterized by the following limitation: the total number of SRs in the pharmacovigilance database is not accompanied by the data on the real level of drug consumption in the population, so we do not know the real population of medication users and are unable to calculate the incidence of SCARs in the total population.

To identify potential safety signals, disproportionality analysis was conducted using ROR and PRR. In pharmacovigilance studies, ROR quantifies the strength of association between a drug and AE by comparing the reporting frequency of a specific drug–AE pair to that of other drug–AE pairs. A ROR value greater than 1 indicates an increased reporting rate for the association. PRR compares the proportion of a particular AE associated with a drug to the proportion of the same AE reported for all other drugs in the database. A PRR value exceeding 1, together with a lower bound of the 95% CI above 1, suggests disproportion in reporting of the AE for the drug, signaling a potential safety concern requiring further investigation.

For disproportionality analysis, we used a 2 × 2 contingency table (Table 9).

The PRR was calculated according to the following formula:(1)$$PRR=\frac{\frac{a}{a+b}}{\frac{c}{c+d}}$$

The 95% CI for the PRR was calculated using the following equation:(2)$$95\%{CI}_{PRR}=e^{\ln\left(PRR\right)\pm 1.96*\sqrt{\frac{1-\frac{a}{a+b}}{a}+\frac{1-\frac{c}{c+d}}{c}}}$$

A signal was considered detected when the following criteria were satisfied: a PRR of at least 2, a chi-squared (χ^2^) statistic of 4 or greater, a minimum of three cases (a ≥ 3), and a lower limit of the 95% confidence interval exceeding 1 [70,71].

The ROR was calculated as the odds of a specific ADR occurring with the suspected drug relative to the odds of the same reaction occurring with all other drugs:(3)$$ROR=\frac{a*d}{b*c}$$

The 95% CI for the ROR was calculated using Woolf’s method:(4)$$95\%CI=e^{\ln\left(ROR\right)\pm 1.96*\sqrt{\frac{1}{a}+\frac{1}{b}+\frac{1}{c}+\frac{1}{d}}}$$

A disproportionality safety signal (association between a specific drug and SCARs) was considered detected if the lower bound of the 95% CI for the ROR exceeded 1, and if there were at least three individual case reports for the drug–event combination of interest (a ≥ 3) [72].

We used the following definitions in our study [73]:

“Adverse reaction—A response to a medicinal product, which is noxious and un-intended. Adverse reaction may arise from use of the product within or outside the terms of the marketing authorization or from occupational exposure. Use outside the marketing authorization includes off-label use, overdose, misuse, abuse, and medication errors.”

“Causality—In accordance with ICH-E2A, the definition of an adverse reaction im-plies at least a reasonable possibility of a causal relationship between a suspected medicinal product and an adverse event. An adverse reaction, in contrast to an adverse event, is characterized by the fact that a causal relationship between a medicinal product and an occurrence is suspected. For regulatory reporting purposes, as detailed in ICH-E2D, if an event is spontaneously reported, even if the relationship is unknown or unstated, it meets the definition of an adverse reaction. Therefore, all spontaneous reports notified by healthcare professionals or consumers are considered suspected adverse re-actions, since they convey the suspicions of the primary sources, unless the reporters specifically state that they believe the events to be unrelated or that a causal relationship can be excluded.”

“A SR is an unsolicited communication by a healthcare professional, or consumer to a competent authority, marketing authorisation holder or other organization (e.g., regional pharmacovigilance center, poison control center) that describes one or more suspected adverse reactions in a patient who was given one or more medicinal products. It does not derive from a study or any organized data collection systems.”

## 5. Conclusions

Studies utilizing data from pharmacovigilance databases represent a valuable tool to analyze real-world data on drug safety. AIS database analysis demonstrated significant contribution of antibacterial agents and MAbs in SCAR developments. Among the top 10 drugs with the highest frequencies of reporting, the strongest signal suggesting an association with SCARs was detected for the antidiabetic medication linagliptin, indicating the need for subsequent studies aimed at the development of complex measures to improve outcomes and enhance clinical practice.

## Figures and Tables

**Figure 1 pharmaceuticals-19-00021-f001:**
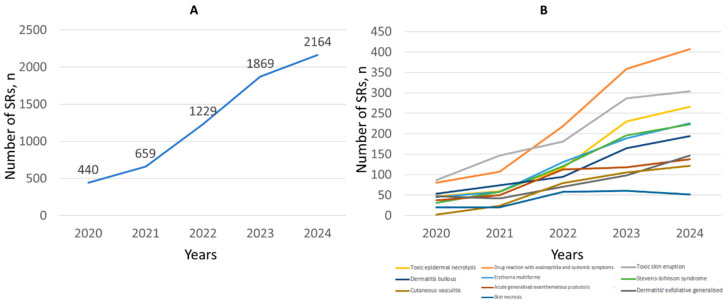
Annual distribution of SRs for SMQ SCAR (**A**) and top 10 PTs (**B**).

**Figure 2 pharmaceuticals-19-00021-f002:**
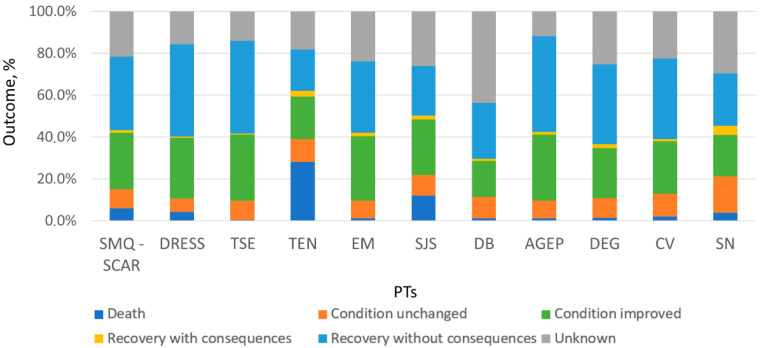
Outcome distribution for the total sample of SRs with SMQ SCAR and for the top 10 PTs. SMQ SCAR—standardized MedDRA query “severe cutaneous adverse reactions”; DRESS—drug reaction with eosinophilia and systemic symptoms; TSE—toxic skin eruption; TEN—toxic epidermal necrolysis; EM—erythema multiforme; SJS—Stevens–Johnson syndrome; DB—dermatitis bullous; AGEP—acute generalized exanthematous pustulosis; DEG—dermatitis exfoliative generalized; CV—cutaneous vasculitis; SN—skin necrosis.

**Figure 3 pharmaceuticals-19-00021-f003:**
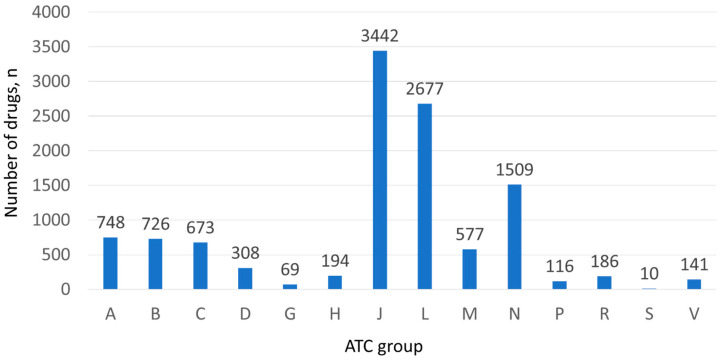
ATC first-level groups involved in SMQ SCAR. A—alimentary tract and metabolism; B—blood and blood-forming organs; C—cardiovascular system; D—dermatologicals; G—genito-urinary system and sex hormones; H—systemic hormonal preparations, excl. sex hormones and insulins; J—antiinfectives for systemic use; L—antineoplastic and immunomodulating agents; M—musculo-skeletal system; N—nervous system; P—antiparasitic products, insecticides, and repellents; R—respiratory system; S—sensory organs; V—various.

**Figure 4 pharmaceuticals-19-00021-f004:**
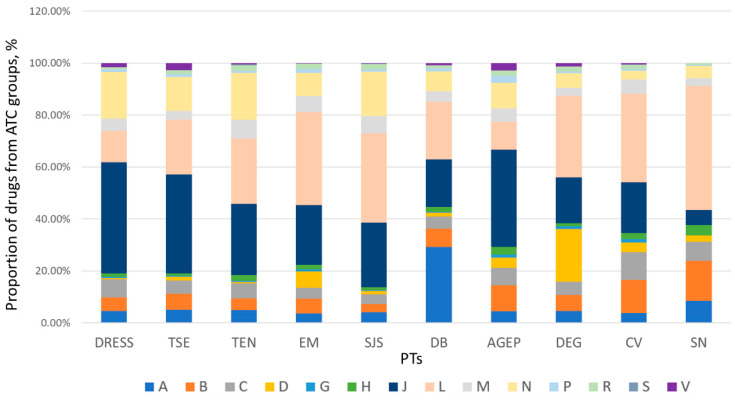
Structure of ATC groups (1st level) revealed in top 10 PTs. DRESS—drug reaction with eosinophilia and systemic symptoms; TSE—toxic skin eruption; TEN—toxic epidermal necrolysis; EM—erythema multiforme; SJS—Stevens–Johnson syndrome; DB—dermatitis bullous; AGEP—acute generalized exanthematous pustulosis; DEG—dermatitis exfoliative generalized; CV—cutaneous vasculitis; SN—skin necrosis.

**Figure 5 pharmaceuticals-19-00021-f005:**
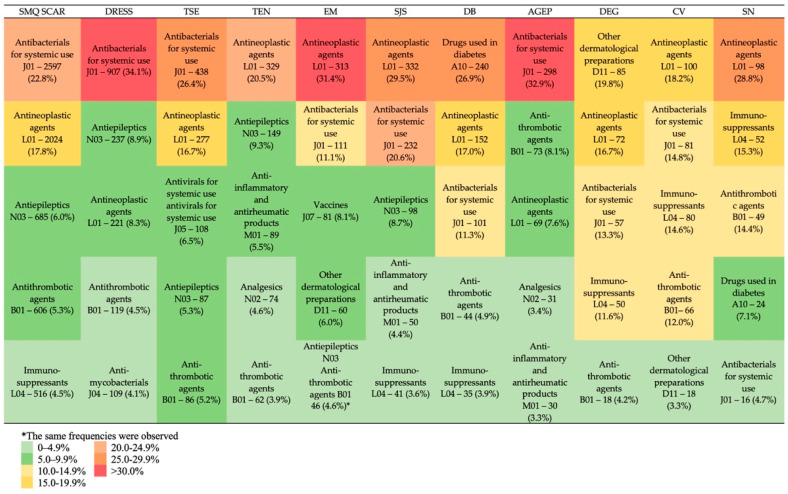
Top 5 ATC second-level groups for SMQ SCAR and top 10 PTs. SMQ SCAR—standardized MedDRA query “severe cutaneous adverse reactions”; DRESS—drug reaction with eosinophilia and systemic symptoms; TSE—Toxic skin eruption; TEN—Toxic epidermal necrolysis; EM—Erythema multiforme; SJS—Stevens-Johnson syndrome; DB—Dermatitis bullous; AGEP—Acute generalized exanthematous pustulosis; DEG—Dermatitis exfoliative generalized; CV—Cutaneous vasculitis; SN—Skin necrosis.

**Figure 6 pharmaceuticals-19-00021-f006:**
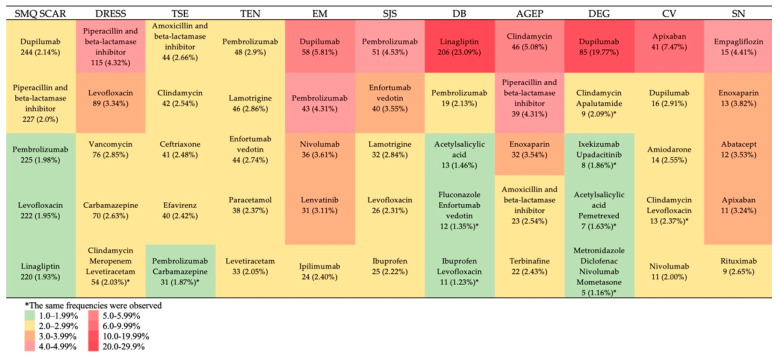
Top 5 drugs with the highest reporting proportions. SMQ SCAR—standardized MedDRA query “severe cutaneous adverse reactions”; DRESS—drug reaction with eosinophilia and systemic symptoms; TSE—toxic skin eruption; TEN—toxic epidermal necrolysis; EM—erythema multiforme; SSJ—Stevens–Johnson syndrome; DB—dermatitis bullous; AGEP—acute generalized exanthematous pustulosis; DEG—dermatitis exfoliative generalized; CV—cutaneous vasculitis; SN—skin necrosis.

**Figure 7 pharmaceuticals-19-00021-f007:**
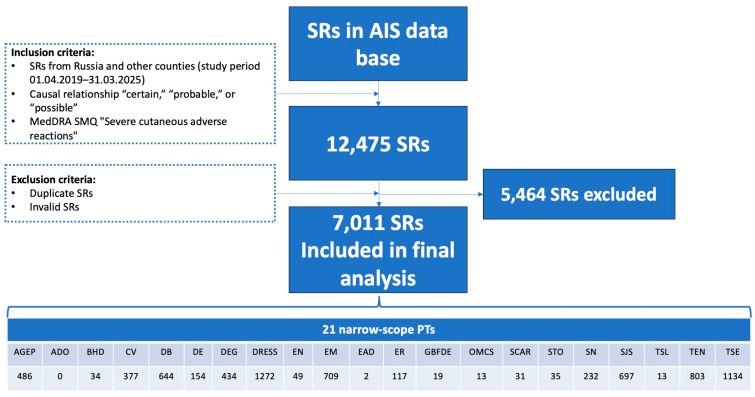
Flowchart of SR selection from the AIS “pharmacovigilance” center. SMQ—standardized MedDRA query; PT—preferred term; SRs—spontaneous reports; AGEP—acute generalized exanthematous pustulosis; ADO—AGEP–DRESS overlap; BHD—bullous hemorrhagic dermatosis; CV—cutaneous vasculitis; DB—dermatitis bullous; DE—dermatitis exfoliative; DEG—dermatitis exfoliative generalized; DRESS—drug reaction with eosinophilia and systemic symptoms; EN—epidermal necrosis; EM—erythema multiforme; EAD—erythrodermic atopic dermatitis; ER—exfoliative rash; GBFDE—generalized bullous fixed drug eruption; OMCS—oculomucocutaneous syndrome; SCAR—severe cutaneous adverse reaction; STO—SJS–TEN overlap; SN—skin necrosis; SJS—Stevens–Johnson syndrome; TSL—target skin lesion; TEN—toxic epidermal necrolysis; TSE—toxic skin eruption.

**Table 1 pharmaceuticals-19-00021-t001:** Structure of PTs within SMQ SCAR and demographic characteristics.

PT	N	Mean Age, Years ± SD	Min	Max, Years	Males, *n* (%)	Females, *n* (%)	Unknown, *n* (%)
AGEP	486	58.8 ± 18.9	10 days	94	288 (59.3)	181 (37.2)	17 (3.5)
AGEP–DRESS overlap	0						
Bullous hemorrhagic dermatosis	34	68.9 ± 19.9	14 years	93	17 (50)	17 (50)	0
Cutaneous vasculitis	377	62.8 ± 19.2	9 years	98	190 (50.4)	167 (44.3)	20 (5.3)
Dermatitis bullous	644	58.67 ± 21.4	5 months	94	294 (45.7)	334 (51.9)	16 (2.5)
Dermatitis exfoliative	154	61.2 ± 21.7	1 year	95	55 (35.7)	86 (55.8)	13 (8.4)
Dermatitis exfoliative generalized	434	59.2 ± 19.8	2 months	97	181 (41.7)	226 (52.1)	27 (6.2)
DRESS	1272	51.3 ± 23.5	2 months	99	615 (48.3)	613 (48.2)	44 (3.5)
Epidermal necrosis	49	55.2 ± 20.0	5 years	92	32 (65.3)	17 (34.7)	0
Erythema multiforme	709	50.9 ± 24.5	1 month	94	371 (52.3)	291 (41.0)	47 (6.6)
Erythrodermic atopic dermatitis	2	69.0 ± 0.0	-	-	0	1 (50)	1 (50)
Exfoliative rash	117	59.3 ± 23.2	2 months	86	63 (53.8)	52 (44.4)	2 (1.7)
Generalized bullous fixed drug eruption	19	50.4 ± 25.9	13 years	88	10 (52.6)	9 (47.4)	0
Oculomucocutaneous syndrome	13	71.3 ± 13.2	51 years	88	0	0	13 (100)
Severe cutaneous adverse reaction	31	50.3 ± 23.6	4 months	84	14 (45.2)	11 (35.5)	6 (19.4)
SJS–TEN overlap	35	46.1 ± 25.1	2 years	85	18 (51.4)	15 (42.9)	2 (5.7)
Skin necrosis	232	59.6 ± 20.7	1 month	92	135 (58.2)	86 (37.1)	11 (4.7)
SJS	697	55.4 ± 22.4	1 month	97	358 (51.4)	291 (41.8)	48 (6.9)
Target skin lesion	13	43.2 ± 25.6	2 months	84	8 (61.5)	5 (38.5)	0
TEN	803	54.3 ± 23.5	3 months	98	407 (50.7)	357 (44.5)	39 (4.9)
Toxic skin eruption	1134	55.6 ± 21.0	6 days	94	634 (55.9)	464 (40.9)	36 (3.2)

**Table 2 pharmaceuticals-19-00021-t002:** Age distribution for the total sample of patients with SCARs and with top 10 PTs.

Age Group (Years)	SMQ—SCAR *n* (%) Total Number—7011	DRESS *n* (%) Total Number—1272	TSE *n* (%) Total Number—1134	TEN *n* (%) Total Number—803	EM *n* (%) Total Number—709	SJS *n* (%) Total Number—697	DB *n* (%) Total Number—644	AGEP *n* (%) Total Number—486	DEG *n* (%) Total Number—434	CV *n* (%) Total Number—377	SN *n* (%) Total Number—232
0–1	53 (0.8)	6 (0.5)	4 (0.4)	4 (0.5)	19 (2.7)	6 (0.9)	3 (0.5)	1 (0.2)	3 (0.7)	0	6 (2.6)
>1–5	123 (1.8)	24 (1.9)	14 (1.2)	21 (2.6)	31 (4.4)	10 (1.4)	14 (2.2)	1 (0.2)	3 (0.7)	0	2 (0.9)
6–11	148 (2.1)	64 (5.0)	16 (1.4)	28 (3.5)	12 (1.7)	12 (1.7)	4 (0.6)	3 (0.6)	4 (0.9)	2 (0.5)	1 (0.4)
12–17	189 (2.7)	58 (4.6)	22 (1.9)	36 (4.5)	20 (2.8)	18 (2.6)	12 (1.9)	9 (1.9)	7 (1.6)	6 (1.6)	1 (0.4)
18–59	2383 (34.0)	492 (38.7)	478 (42.2)	262 (32.6)	243 (34.3)	215 (30.8)	154 (23.9)	189 (38.9)	141 (32.5)	108 (28.6)	66 (28.4)
60–74	1815 (25.9)	330 (25.9)	307 (27.1)	215 (26.8)	158 (22.3)	160 (23.0)	179 (27.8)	155 (31.9)	105 (24.2)	109 (28.9)	65 (28.0)
75–89	1165 (16.6)	173 (13.6)	176 (15.5)	142 (17.7)	87 (12.3)	97 (13.9)	145 (22.5)	85 (17.5)	90 (20.7)	83 (22.0)	42 (18.1)
≥90	124 (1.8)	15 (1.2)	19 (1.7)	8 (1.0)	9 (1.3)	12 (1.7)	18 (2.8)	6 (1.2)	10 (2.3)	14 (3.7)	4 (1.7)
No data	1011 (14.4)	110 (8.6)	98 (8.6)	87 (10.8)	130 (18.3)	167 (24.0)	115 (17.9)	37 (7.6)	71 (16.4)	55 (14.6)	45 (19.4)

SMQ SCAR—standardized MedDRA query “severe cutaneous adverse reactions”; DRESS—drug reaction with eosinophilia and systemic symptoms; TSE—toxic skin eruption; TEN—toxic epidermal necrolysis; EM—erythema multiforme; SJS—Stevens–Johnson syndrome; DB—dermatitis bullous; AGEP—acute generalized exanthematous pustulosis; DEG—dermatitis exfoliative generalized; CV—cutaneous vasculitis; SN—skin necrosis.

**Table 3 pharmaceuticals-19-00021-t003:** SR distribution by country of origin.

Country	*n*	%
France	1560	22.1
Russia	985	14.1
Japan	603	8.6
USA	428	6.2
China	417	6.1
Spain	297	4.2
Singapore	241	3.4
Italy	218	3.1
Canada	197	2.8
United Kingdom	175	2.5
Germany	161	2.4
Portugal	152	2.1
Switzerland	74	1.3
Poland	73	1.0
Australia	72	1.0
South Korea	71	1.0
Sweden	69	1.0
Netherlands	62	0.9
India	54	0.8
Other *	712	9.9
No data	390	5.5

* Countries with <50 SRs.

**Table 4 pharmaceuticals-19-00021-t004:** Structure of reporters.

Reporter	*n* (Total—7011)	%
Pharmaceutical company	5974	85.2
Healthcare professional	701	10.0
Regional pharmacovigilance center	136	1.9
Other (e.g., distributor or other organization)	133	1.9
Competent authority	64	0.9
WHO Collaborating Centers for International Drug Monitoring	1	0.01
Unknown	2	0.03

**Table 5 pharmaceuticals-19-00021-t005:** SR distribution based on the seriousness criteria.

Seriousness Criteria	*n* (Total—7011)	%
Death	407	5.8
Life-threatening	240	3.4
Life-threatening; hospitalization or prolongation of existing hospitalization	353	5.1
Life-threatening; hospitalization or prolongation of existing hospitalization; disability/incapacity	14	0.2
Life-threatening; disability or permanent damage	2	0.03
Disability/incapacity	34	0.5
Hospitalization or prolongation of existing hospitalization	2761	39.6
Hospitalization or prolongation of existing hospitalization; disability/incapacity	43	0.6
Important medical events	3110	44.4

**Table 6 pharmaceuticals-19-00021-t006:** Signal detection using PRR for the top 10 drugs with the highest reporting frequencies at the SMQ SCAR level.

Drug	PRR	CI 95%
Dupilumab	1.20	1.06–1.36
Piperacillin and beta-lactamase inhibitor	10.07	8.86–11.43
Pembrolizumab	1.67	1.47–1.91
Levofloxacin	4.14	3.63–4.72
Linagliptin	15.37	13.54–17.44
Clindamycin	12.44	10.89–14.21
Carbamazepine	8.64	7.50–9.96
Ceftriaxone	2.07	1.78–2.40
Nivolumab	1.85	1.58–2.16
Amoxicillin and beta-lactamase inhibitor	4.47	3.81–5.25

**Table 7 pharmaceuticals-19-00021-t007:** Signal detection using ROR for the top 10 drugs with the highest reporting frequencies at the SMQ SCAR level.

Drug	ROR	CI 95%
Dupilumab	1.20	1.06–1.37
Piperacillin and beta-lactamase inhibitor	10.81	9.42–12.40
Pembrolizumab	1.68	1.47–1.92
Levofloxacin	4.24	3.70–4.86
Linagliptin	17.24	14.95–19.88
Clindamycin	13.62	11.77–15.77
Carbamazepine	9.17	7.88–10.68
Ceftriaxone	2.08	1.79–2.43
Nivolumab	1.86	1.59–2.18
Amoxicillin and beta-lactamase inhibitor	4.59	3.89–5.42

**Table 8 pharmaceuticals-19-00021-t008:** “Narrow” PTs within SMQ SCAR used in our study.

PT	MedDRA Code
Acute generalized exanthematous pustulosis	10048799
AGEP-DRESS overlap	10089003
Bullous hemorrhagic dermatosis	10083809
Cutaneous vasculitis	10011686
Dermatitis bullous	10012441
Dermatitis exfoliative	10012455
Dermatitis exfoliative generalized	10012456
DRESS	10073508
Epidermal necrosis	10059284
Erythema multiforme	10015218
Erythrodermic atopic dermatitis	10082985
Exfoliative rash	10064579
Generalized bullous fixed drug eruption	10084905
Oculomucocutaneous syndrome	10030081
Severe cutaneous adverse reaction	10085778
SJS-TEN overlap	10083164
Skin necrosis	10040893
SJS	10042033
Target skin lesion	10081998
TEN	10044223
Toxic skin eruption	10057970
SMQ—Severe cutaneous adverse reactions ^1^	20000020

^1^ This is an SMQ-level term that includes all 21 narrow-scope PTs.

**Table 9 pharmaceuticals-19-00021-t009:** Two-by-two contingency table.

	Reaction of Interest	All Other Reactions
Suspected drug	a	b
All other drugs	c	d

where a = number of reports for the selected drug with SCARs; b = number of reports for the selected drug with any other ADRs; c = number of reports for all other drugs with SCARs; d = number of reports for all other drugs with any other ADRs.

## Data Availability

The original contributions presented in this study are included in the article/Appendix A. Further inquiries can be directed to the corresponding author.

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
