# Peer review of "Real-World Data on Severe Cutaneous Adverse Reactions to Drugs"

_pharmaceuticals, 2025, doi:10.3390/ph19010021_

Round 1

Reviewer 1 Report

Comments and Suggestions for Authors

It was my pleasure to be appointed to review the research article entitled “real world data on severe cutaneous adverse reaction to drugs” by Sergey Zyryanov and co-authors. The article is offering a valuable and well-structured discussion on the structure of medications implicated in SCARs and examines drug-induced SCAR signals by computing the ROR and PRR using SRs taken from the Russian National Pharmacovigilance database.

However, there are some comments/suggestion that may be incorporated.

  1. Table no. 1, AGEP-DRESS overlap, no case found or data were unviable?
  2. Table no. 2, how the age groups were decided?
  3. Table no. 4, what was the reason for low number found with WHO Collaborating Centers for International Drug Monitoring?
  4. In table no. 5, the total-7011 may be taken in brackets.
  5. In table no. 9, the alphabets A and C are capital while b and d are small?

Author Response

Comment # 1: Table no. 1, AGEP-DRESS overlap, no case found or data were unviable?

Response # 1: Thank you for the question! There were no cases of AGEP-DRESS overlap identified in AIS database in our study.  

Comment #2: Table no. 2, how the age groups were decided?

Response #2: Thank you for the question! Age group used in Table 2 were defined based on World Health Organization recommendations (infancy (0-1 year), early childhood (1-5 years), middle childhood (6-11 years), adolescents (12-17 years), combined category of young adults and adults (18-59), elderly/early old (60-74 years), oldest old (75–89 years), longevity (≥90))  

Comment #3: Table no. 4, what was the reason for low number found with WHO Collaborating Centers for International Drug Monitoring?

Response #3: The question is appreciated. The available data do not permit a definitive explanation for the low reporting rates of SCARs by WHO Collaborating Centers for International Drug Monitoring in the AIS database; rather, our findings are limited to characterizing this phenomenon.  

Comment #4:  In table no. 5, the total-7011 may be taken in brackets.

Response #4: Thank you for attention, corresponding changes have been made to the table 5.  

Comment #5: In table no. 9, the alphabets A and C are capital while b and d are small?

Response #5: Thank you for attention, all letters should be small, corresponding changes have been made to the table 9.

Reviewer 2 Report

Comments and Suggestions for Authors

This informative report on "Real-World Data on Severe Cutaneous Adverse Reactions to Drugs" is based on electronic data. Although the topic is of interest the presentation is difficult to follow. It will be beneficial to subdivide data under sub-headlines and minimize use of acronyms (e.g. what is SMQ?) .

In the same line - some figure are difficult to read with very small font e.g. Figure 5 , It also beneficial to add a headline to each figure/table.

What re the differences between Figure 4 and 5 ?

Author Response

Comment #1: This informative report on "Real-World Data on Severe Cutaneous Adverse Reactions to Drugs" is based on electronic data. Although the topic is of interest the presentation is difficult to follow. It will be beneficial to subdivide data under sub-headlines and minimize use of acronyms (e.g. what is SMQ?) .

Response #1: Thank you for the work done to review our manuscript. Using your recommendations we added sub-headlines in the section Results (2.1 General characteristics of SMQ SCAR; 2.2 Suspected drugs; 2.3 Disproportionality analysis). Abbreviations used in our manuscript are mainly presented in tables and figures, and full translation is given in footnotes following each table and figure.  SMQ is abbreviation of Standardized Medical Dictionary for Drug Regulatory Activities Query, it was first indicated on page 3 just in the beginning of Results section.

Comment #2: In the same line - some figure are difficult to read with very small font e.g. Figure 5 , It also beneficial to add a headline to each figure/table.

Response #2: We appreciate your recommendations and tried to enlarge the font, but within the manuscript figures are anyway small, that is limited by the page size. Readability is optimal when opening the figure in full size, that is the option realized only in online version of a journal.

Comment #3: What re the differences between Figure 4 and 5 ?

Response #3: Thank you for the question, Figure 4 demonstrates 1st level ATC groups revealed in in top-10 preferred terms, while Figure 5 contains data on top-5 2nd level ATC groups for SMQ SCAR and top-10 preferred tterms.

Reviewer 3 Report

Comments and Suggestions for Authors

The manuscript addresses a relevant and increasingly important topic: SCARs (Severe Cutaneous Adverse Reactions) analysed using a national pharmacovigilance database. The large number of reports and the real-world approach add value to the study. However, the manuscript presents conceptual, methodological, and clarity issues. Substantial revisions are required to ensure scientific rigour, terminological consistency, and methodological robustness.

The authors use the terms severity and seriousness interchangeably, which is incorrect in pharmacovigilance. Seriousness refers to regulatory criteria (hospitalisation, life-threatening condition, ….), while severity refers to the clinical intensity of symptoms. In the manuscript, the authors apply seriousness criteria but repeatedly refer to them as “severity”. It is essential to clarify the concepts and correct the terminology throughout the text.

Why was the Naranjo algorithm considered?
The authors should justify the choice of the Naranjo algorithm and clearly discuss its limitations.

According to the WHO definition, “possible” implies that the reaction may be explained either by the drug or by other clinical conditions and/or concomitant medications. In SCARs, where the differential diagnosis is broad, this category introduces significant uncertainty. The authors should justify the inclusion of such cases and ideally perform a separate analysis excluding “possible”.

It is essential to explain:
• why only the Russian database was used,
• whether there are particular characteristics of the Russian reporting system that may introduce bias,
• the impact of the high proportion of industry reports (≈85%) on reporting bias,
• how cases involving multiple suspected medicines were managed.
Without this clarification, the interpretative strength of the results is compromised.

The discussion section repeats many results, and it is important to reinforce the study limitations. PRR/ROR identify signals, not causality.

There is excessive use of abbreviations without definition in table footnotes.
Figures also contain too much information, making them difficult to interpret.
Simplification is recommended.

The introduction and methodological description contain overly long explanations of epidemiological and statistical concepts that are already well established. Reducing these sections would improve focus and readability.

Comments on the Quality of English Language

The English could be improved 

Author Response

Comment #1: The authors use the terms severity and seriousness interchangeably, which is incorrect in pharmacovigilance. Seriousness refers to regulatory criteria (hospitalisation, life-threatening condition, ….), while severity refers to the clinical intensity of symptoms. In the manuscript, the authors apply seriousness criteria but repeatedly refer to them as “severity”. It is essential to clarify the concepts and correct the terminology throughout the text.

Response #1: Thank you for your comments. Indeed, when discussing the severity of an adverse reaction, the term "seriousness" should be used. Changes have been made in text of manuscript regarding only seriousness consideration.

Comment #2: Why was the Naranjo algorithm considered?
The authors should justify the choice of the Naranjo algorithm and clearly discuss its limitations.

Response #2: Thank you for comments! The Naranjo Algorithm is a validated and structured method used for over 40 years in routine pharmacovigilance to assess the causal relationship between a suspected medicinal product and an adverse drug reaction. Currently, the Naranjo Algorithm is considered the "gold standard" for the initial assessment of ADR cases in clinical trials and is one of the most well-known and frequently used algorithms worldwide, allowing for the comparison of data from different authors and studies. Its advantages include standardization (common criteria, reproducibility), objectivity (minimizes subjectivity and variability in the assessment of the ADR-drug causal relationship), and simplicity and speed of use. Limitations include low sensitivity in complex and non-standard cases. In such cases, the issue of causal relationship was reviewed collegially by clinicians and expert staff of the Pharmacovigilance Center (at least 3 different persons).

Comment #3: According to the WHO definition, “possible” implies that the reaction may be explained either by the drug or by other clinical conditions and/or concomitant medications. In SCARs, where the differential diagnosis is broad, this category introduces significant uncertainty. The authors should justify the inclusion of such cases and ideally perform a separate analysis excluding “possible”.

Response #3: Thank you for comments! Our study was aimed on identification of a list of drugs involved in SCARs development. "Certain," "probable/likely," and "possible" causal relationships are considered in the pharmacovigilance system as high levels of certainty in causality assessment, thus SRs in which such levels of causality were identified were considered relevant for the aims of our study. In our study, we deliberately excluded cases with " Unlikely", “Conditional / Unclassified”, and “Un-assessable / Unclassifiable” relationships.

Comment #4: It is essential to explain: why only the Russian database was used

Response #4: Thank you for comments! The present study utilized data obtained from Roszdravnadzor, the Russian authority responsible for the collection of ADRs. The analysis of national pharmacovigilance databases is a common practice among researchers worldwide, as evidenced by numerous publications reporting findings from databases such as the FDA Adverse Event Reporting System (FAERS) in the United States, the Japanese Adverse Drug Event Report database (JADER), the Base Nationale de Pharmacovigilance (BNPV) in France, etc. Our research team offers a comprehensive characterization of the spontaneous reports collected in the Roszdravnadzor AIS database, thereby enabling a detailed examination of SCARs.

Comment #5: It is essential to explain: whether there are particular characteristics of the Russian reporting system that may introduce bias

Response #5: Thank you for comments! As outlined in the "Materials and Methods" section, Automatized Information System (AIS) "Pharmacovigilance" operates in accordance with the ICH E2B (R3) standard. It fully adheres to international standards for pharmacovigilance databases. The "Eurasian Economic Union (EAEU) GVP," whose most recent version was adopted in 2022, is implemented within the Russian Federation. This "EAEU GVP" is aligned with the European Medicines Agency's "EMA GVP" as of early 2021. Consequently, there are no substantive differences between the Russian reporting framework and those employed in the EU and globally, thereby minimizing potential bias in data collection and analysis.

Comment #6: : It is essential to explain: the impact of the high proportion of industry reports (≈85%) on reporting bias

Response #6: Thank you for comments!  According to regulatory documents, primary responsibility for the efficacy and safety of relevant medicinal products in the Russian Federation lies with their marketing authorization holders/manufacturers. Healthcare professionals and patients often report adverse reactions not directly to the regulator, but through the marketing authorization holders/manufacturers, which largely explains the high proportion of industry reports. Furthermore, marketing authorization holders operating in the Russian Federation are required to report adverse reactions to their medicinal products to regulatory authorities, not only those occurring in the Russian Federation but also those occurring abroad. It is highly likely that the company also receives such reports in other countries from healthcare professionals. A comparison of information received from healthcare professionals and the marketing authorization holders revealed no significant differences in the quality of individual case safety reports. Therefore, in our opinion, the high proportion of industry reports does not affect the quality of our spontaneous reporting data.

Comment #7: It is essential to explain: how cases involving multiple suspected medicines were managed.

Response #7: Thank you for comments! If a single report included multiple suspected drugs, the causal relationship was assessed for each drug independently by several experts (pharmacovigilance experts and clinical pharmacologists). Only drugs demonstrating a high level of association (at least possible) were included in the final analysis.

Comment #8: The discussion section repeats many results, and it is important to reinforce the study limitations. PRR/ROR identify signals, not causality.

Response #8: Thank you for comments! Data on limitations was added in the closing of Discussion section:

“Finally, we need to make an emphasis on limitations of our study which may impact the interpretation and reliability of our findings. Despite the method of SRs represents the cornerstone of the postmarketing surveillance of drug safety, the nature of SRs may lead to underreporting and selective reporting bias, limiting representativeness and comprehensive safety profiling. The most salient limitation lies in the fact that the cumulative number of SRs does not correspond to the actual consumption rates of a given drug in the population, thereby precluding the estimation of true ADRs incidence. It should be added that retrospective character of our analysis, combined with the potential for incomplete data indicated in SRs limits the ability to accurately predict the risk factors associated with the development of SCARs. Another important concern is related to disproportionality analysis (ROR and PRR calculation). Disproportionality analysis is aimed on identification of patterns of disproportionate reporting of specific AEs in patients exposed to a particular drug, relative to overall reporting rates within pharmacovigilance database. However, these statistical disproportions do not establish direct causal relationships. The observed strength of a signal may arise from alternative explanations summarized in the work by Fusaroli M et al (2025) [65]: reverse causality (drug was used to treat the event after diagnosis), protopathic bias (drug was used to address early symptoms of an undiagnosed event), indication bias (underlying condition prompting drug use increases susceptibility to the event), notoriety bias (heightened regulatory or media attention enhances reporting), selective reporting practices, and variability among reporters.”

Comment #9: There is excessive use of abbreviations without definition in table footnotes.
Figures also contain too much information, making them difficult to interpret.
Simplification is recommended.

Response # 9: Thank you for comments! Abbreviations often mentioned in the main text of manuscript (e.g.: SRs – spontaneous reports; PTs – preferred terms) were given in Tables and Figures without translation, while all other abbreviations (e.g: different forms of SCARs, different PTs) are given with full names after each table and each figure in footnotes.

Comment #10: The introduction and methodological description contain overly long explanations of epidemiological and statistical concepts that are already well established. Reducing these sections would improve focus and readability.

Response #10: Thank you for comments. We suppose that a profound description of a problem in introduction will contribute to a better understanding of a complex SCAR problem by readers. Detailed methodological description is aimed on elimination of possible questions regarding quality of our study.

Round 2

Reviewer 2 Report

Comments and Suggestions for Authors

Font in figures is too small to read

Author Response

Comment #1: Font in figures is too small to read Response#1: Thank you for the comments, we have changed Figures as far as it was possible. Thi final view of figures will be different since opening of the figure on the site of journal will result in a full screen size.

Reviewer 3 Report

Comments and Suggestions for Authors

Dear Authors,

Thank you for your detailed responses. I appreciate the effort to address the points raised; however, a few clarifications and adjustments are still necessary. Several of the responses you provided contain important methodological justifications (e.g., database selection, handling of multiple suspected drugs, rationale for inclusion of “possible” cases). These clarifications must be incorporated into the manuscript itself so that readers — not only reviewers — can fully understand and assess the study.

For example the Choice of Naranjo Algorithm: The response provides a general justification; however, this explanation must be included in the manuscript itself, not only in the response letter.
The point related with the Inclusion of “Possible” Cases: the justification provided does not fully align with the WHO definition of “possible.”By definition, “possible” implies that the event could also be explained by concomitant diseases or medications, and SCARs have a wide differential diagnosis. Thus, the level of certainty is not “high.” You may certainly choose to include “possible” cases, but the justification should be reframed to reflect this reality.

Additionally, the Use of the Russian Database and Reporting Characteristics: Your explanation is clear, but these important points must also appear in the manuscript.

Comments on the Quality of English Language

The English could be improved 

Author Response

Comment #1: Thank you for your detailed responses. I appreciate the effort to address the points raised; however, a few clarifications and adjustments are still necessary. Several of the responses you provided contain important methodological justifications (e.g., database selection, handling of multiple suspected drugs, rationale for inclusion of “possible” cases). These clarifications must be incorporated into the manuscript itself so that readers — not only reviewers — can fully understand and assess the study. For example the Choice of Naranjo Algorithm: The response provides a general justification; however, this explanation must be included in the manuscript itself, not only in the response letter.

Response #1: Thank you for the work done to improve the quality of our manuscript! We have added explanations of the choice of Naranjo Algorithm in the text of manuscript in the section Materials and Methods.  

Comment #2: The point related with the Inclusion of “Possible” Cases: the justification provided does not fully align with the WHO definition of “possible.” By definition, “possible” implies that the event could also be explained by concomitant diseases or medications, and SCARs have a wide differential diagnosis. Thus, the level of certainty is not “high.” You may certainly choose to include “possible” cases, but the justification should be reframed to reflect this reality.

Response #2: Thank you for the comment! We have added justification of the choice of “possible” causality category in the text of manuscript, section Materials and Methods.  

Comment #3: Additionally, the Use of the Russian Database and Reporting Characteristics: Your explanation is clear, but these important points must also appear in the manuscript.

Response #3: Thank you for the comment! We have added issues regarding choice of Russian Database in the text of manuscript in the Materials and Methods section.